# Does Vitamin D3 Supplementation Improve Depression Scores among Rural Adolescents? A Randomized Controlled Trial

**DOI:** 10.3390/nu16121828

**Published:** 2024-06-11

**Authors:** Pradeep Tarikere Satyanarayana, Ravishankar Suryanarayana, Susanna Theophilus Yesupatham, Sudha Reddy Varadapuram Ramalingareddy, Navya Aswathareddy Gopalli

**Affiliations:** 1Community Medicine, Sri Devaraj Urs Medical College (SDUMC) Sri Devaraj Urs Academy of Higher Education and Research (SDUAHER), Kolar 563103, India; 2Biostatistics, Sri Devaraj Urs Medical College (SDUMC) Sri Devaraj Urs Academy of Higher Education and Research (SDUAHER), Kolar 563103, India; 3Biochemistry, Sri Devaraj Urs Medical College (SDUMC) Sri Devaraj Urs Academy of Higher Education and Research (SDUAHER), Kolar 563103, India; sussannaty@sduaher.ac.in; 4Pediatrics, Sri Devaraj Urs Medical College (SDUMC) Sri Devaraj Urs Academy of Higher Education and Research (SDUAHER), Kolar 563103, India; 5Psychiatry, Sri Devaraj Urs Medical College (SDUMC) Sri Devaraj Urs Academy of Higher Education and Research (SDUAHER), Kolar 563103, India

**Keywords:** vitamin D supplementation, rural adolescents, depression, RCT

## Abstract

Background: Contemporary evidence has been established demonstrating that stunted vitamin D levels are associated with depression, poor mood, and other mental disorders. Individuals with normal vitamin D levels have a much lower probability of developing depression. Improving vitamin D levels by supplementation has shown betterment in depressive patients among different age groups. The objective of this study was to assess the effect of vitamin D supplementation on depression scores among rural adolescents. Material and methods: This study was a cluster randomized controlled trial carried out for a period of 3 years among adolescents from rural Kolar. The sample size was calculated based on previous research and was determined to be 150 for each group. The intervention arm received 2250 IU of vitamin D, and the control arm received a lower dose of 250 IU of vitamin D for 9 weeks. To assess sociodemographic status, a pretested, semi-structured questionnaire was used, and, to assess depression, the Beck Depression Inventory (BDI-II) was used. A baseline assessment was carried out for vitamin D status and depression status, followed by a post-intervention assessment. From the start of the trial, the participants were contacted every week by the pediatric team to investigate any side effects. Results: Out of 235 school students in the vitamin D supplementation arm, 129 (54.9%) belonged to the 15 years age group, 124 (52.8%) were boys, and 187 (79.6%) belonged to a nuclear family. Out of 216 school students in the calcium supplementation arm, 143 (66.2%) belonged to the 15 years age group, 116 (53.7%) were girls, and 136 (63%) belonged to a nuclear family. By comparing Beck depression scores before and after the intervention, it was found that the vitamin D intervention arm showed a statistically significant reduction in Beck depression scores. Conclusions: The present study showed that vitamin D supplementation reduced depression scores, showing some evidence that nutritional interventions for mental health issues such as depression are an excellent option. Vitamin D supplementation in schools can have numerous beneficiary effects on health while mutually benefiting mental health.

## 1. Introduction

The World Health Organization (WHO) has clearly stated that depression is a debilitating illness with innumerable causes. Social, environmental, genetic, and psychological determinants have been extensively studied worldwide as possible risk factors; however, the roles of diet and nutrients remain little explored. Causal associations between dietary patterns and depression propound that the presence of certain nutrients within the diet may play a pivotal role in perpetuating good mental health, while nutritional deficiencies may have an appreciable impact on mood and overall wellbeing [1]. Depression is a mental illness that can affect the quality of life of the affected individual. According to the DSM V there are nine symptoms, of which the presence of only five can be diagnostic, attracting attention for immediate intervention. Depression has many well-established treatments, such as pharmacological interventions, cognitive behavioral therapy, interpersonal and psychodynamic therapy, and sometimes electroconvulsive therapy (ECT). However, newer theories suggest that micronutrient deficiencies can also lead to depression, of which vitamin D deficiency is a forerunner [2]. With increasing research providing evidence on the relationship between vitamin D and depression, many possible causative roles have been established. As a neuroactive steroid, vitamin D plays a crucial role in neurotransmitter regulation and neuroimmunomodulation [3]. Specific brain regions related to depression, such as the prefrontal cortex, hippocampus, cingulate gyrus, thalamus, hypothalamus, and substantia nigra, have vitamin D receptors, suggesting that vitamin D may possibly be related to the pathophysiology of depression [4]. Vitamin D also modulates the hypothalamic–pituitary–adrenal axis, which, in turn, regulates the production of the monoamine neurotransmitters epinephrine, norepinephrine, and dopamine in the adrenal cortex and protects against the depletion of dopamine and serotonin, directly establishing a possible causal association between depression and vitamin D levels. Active vitamin D also has anti-inflammatory properties, which may counter the increase in inflammatory cytokines associated with depression [5]. A critical appraisal showed that vitamin D supplementation can be extremely beneficial in treating depression, as depression is related to vitamin D levels [6,7]. Vitamin D supplementation has a potential therapeutic role in anxiety and depression because of its antioxidant, anti-inflammatory, pro-neurogenic, and neuromodulatory properties, which have already been sufficiently demonstrated [8].

In a tropical country such as India, vitamin D deficiency is still a public health problem despite the bountiful sunshine, which is a potential source of vitamin D [9]. There could be many reasons for vitamin D deficiency in a country such as India—for example, race, ethnicity, sunlight exposure, skin color, and cooking practices [10]. A systematic review showed that vitamin D is crucial, especially when the central nervous system of the fetus develops, and that the brain is sensitive to maternal nutritional deficiencies [11]. A study conducted by Williams et al. showed that lower vitamin D levels also predict depression, suggesting a quintessential role of vitamin D in depression [12]. A study conducted in Himachal Pradesh, India, showed that vitamin D deficiency is as high as 80% among adolescents [13]. Vitamin D deficiency is extremely common among particular age groups for many reasons, and, because it is extremely common, it is ignored [14]. There are very few studies of vitamin D deficiency and its possible causal relationships in adolescents. Thus, with this background, we planned the present study with the objective of evaluating the effectiveness of vitamin D supplementation in the treatment of depression in rural adolescent boys and girls.

## 2. Materials and Methods

The present study was a cluster randomized controlled trial carried out for a period of 3 years from January 2021 to December 2023, conducted in rural areas of Kolar (Latitude: 13°07′48.00″ N and Longitude: 78°07′48.00″ E). Kolar is one of the districts in Karnataka, India. Kolar has more than 36 rural schools, out of which, 20 were randomly selected. The sample size was estimated based on the mean difference in depression score determined in a study by Libuda et al., in which an average scale variance score of 13.3 (SD) was obtained. Expecting a reduction of 20% in Beck Depression Inventory scores in the intervention group compared to in the placebo group, with 80% power and an alpha error of 5%, the estimated sample size per group was 96. Expecting a dropout rate of 5% during the study and considering that the design had a cluster design effect of 1.5, the final sample size achieved in each group was 150 [15]. A total of 20 rural schools in Kolar were selected. Each school was considered as a cluster. The study was carried out during the summer months of February and March for the included schools and the geographical location did not differ amongst the schools. Block randomization was performed using the online software randomization.com. There were two groups after randomization: one group received a high dose of vitamin D at 2000 IU per day, and the other received a lower dose of 250 IU per day, along with calcium. All adolescents aged 14–19 years were included in the study after consent and assent were obtained. The adolescent boys and girls were screened, and those with the following were excluded from the study: any pre-existing mental health illnesses such as previously diagnosed severe depression or a history of suicidal tendencies or attempts; previously confirmed renal abnormalities; previously confirmed cardiac abnormalities; and previously confirmed neurological disorders, such as epilepsy. To assess sociodemographic status, a pretested, semi-structured questionnaire was used. To assess depression, the Beck Depression Inventory (BDI-II) was used, which contains 21 items rated on a Likert scale. Participants with any form of depression were included in the study. When using the Beck Depression Inventory scale, the scores for each of the 21 questions are added up. The highest possible score for the whole test is sixty-three, and the lowest possible score is zero. The determination of various categories of depression is based on the summed-up scores: 0–10 means that the mood fluctuations are considered normal, 11–16 indicates minimal depression, 17–20 indicates mild depression, 21–30 indicates moderate depression, 31–40 indicates severe depression, and more than 40 indicates extreme depression. The present study used the cutoff values as per BDI-II, where 0–13 was considered minimal depression, 14–19 mild depression, 20–28 moderate depression, and 29–63 as severe depression [16]. All school children were interviewed by the Associate Professor of the Department of Community Medicine, who had prior experience in using this BDI scale. Venous blood was taken by an experienced lab technician and analyzed by Central Diagnostic Laboratory Services, Biochemistry Department, SDUMC, SDUAHER, Kolar. Baseline vitamin D3 levels were assessed via blood sampling using VITROS Immunodiagnostic products with a 25-OH Total reagent pack. The diagnostic cut-off levels of serum vitamin D (ng/mL) are classified as follows: deficiency < 20 ng/mL, insufficiency 21–29 ng/mL, sufficiency > 30 ng/mL, and toxicity > 150 ng/mL [17]. The present study used the Indian Academy of Pediatrics classification for the diagnostic cut-off of vitamin D levels, in which less than 12 ng/mL is considered deficient, 12–20 ng/mL is considered insufficient, 20–100 ng/mL is considered sufficient, and more than 100 ng/mL is considered toxic [18]. Block randomization was conducted by an expert from the Department of Community Medicine using the free online software randomization.com. Vitamin D3 supplementation was achieved by administering 60,000 IU of vitamin D3 once a month for two months, totaling 2000 IU per day, and by administering 500 mg of calcium with 250 IU of vitamin D3 per day for 9 weeks. A total of 2250 IU of vitamin D was taken by the vitamin D intervention arm. The other arm received 500 mg of calcium in addition to 250 IU of vitamin D3 per day for 9 weeks. Both vitamin D3 and calcium were administered at a safe dose, and the fear of toxicity was minimal. Both tablets were taken in front of the investigating team to ensure adherence. After 9 weeks of intervention, vitamin D3 levels were again assessed using the methods previously described. From the start of the trial, the participants were contacted every week by the pediatric team. A take-part book was given to all participants to examine drug adherence, a medical examination was carried out at the start of the study, follow-up was carried out weekly, and any side effects were documented. When depression was found in a student, even after the trial, they were referred to a psychiatrist. Flow chart of the study participants from recruitment to completion of the trial has been mentioned as in Figure 1. The study commenced after obtaining approval from the Central Ethics Committee (SDUAHER/Res.Proj.173/2020-21). Informed written consent/assent was obtained from the school children after informing them about the benefits and risks of the study. Autonomy was maintained by the study participants, as participation in the trial was voluntary. Confidentiality was also maintained, as the participants’ names and personal details were not recorded. The study was registered in CTRI (REF No/2021/03/042355). All collected data were entered into Microsoft Excel and analyzed using SPSS v 22 (IBM Corp, Armonk, NY, USA). Sociodemographic data are expressed using descriptive statistics, such as frequencies and percentages. Inferential statistics, such as Chi square, were applied to determine associations between variables. Pre-intervention and post-intervention differences were assessed using *t*-tests, summarized as the mean and standard deviation (SD), with a statistically significant difference defined as a *p* value of less than 0.05. Both an intention-to-treat (ITT) analysis and a per-protocol analysis were conducted and are reported separately. *p* < 0.05 was accepted as statistically significant.

## 3. Results

Out of the 235 school students in the vitamin D supplementation arm, 129 (54.9%) belonged to the 15 years age group, 124 (52.8%) were boys, 187 (79.6%) belonged to a nuclear family, 195 (83%) had a mixed diet, 166 (70.6%) engaged in outdoor activities in the afternoon, and 184 (78.3%) engaged in outdoor activities for more than 30 min. Out of the 216 school students in the calcium supplementation arm, 143 (66.2%) belonged to the 15 years age group, 116 (53.7%) were girls, 136 (63%) belonged to a nuclear family, 184 (85.2%) had a mixed diet, 160 (74.1%) engaged in outdoor activities in the evening, and 111 (51.4%) engaged in outdoor activities for more than 30 min (Table 1).

In the vitamin D intervention arm (2000 IU per day), 235 students gave blood for a vitamin D analysis before and after vitamin D supplementation. Before the vitamin D supplementation, 53 (22.6%) belonged to the vitamin D deficiency group, 111 (47.2%) belonged to the vitamin D insufficient group, and 71 (30.2%) belonged to the vitamin D sufficient group. After the vitamin D supplementation, 54 (23%) belonged to the vitamin D deficiency group, 36 (15.3%) belonged to the vitamin D insufficient group, and 145 (61.7%) belonged to the vitamin D sufficient group. In the calcium intervention arm (500g plus 250 IU of vitamin D), 216 students gave blood for a vitamin D analysis before and after the calcium supplementation, and 190 gave blood for a vitamin D supplementation analysis. Before the calcium supplementation, 68 (31.4%) belonged to the vitamin D deficiency group, 59 (27.3%) belonged to the vitamin D insufficient group, and 89 (41.3%) belonged to the vitamin D sufficient group. After the vitamin D supplementation, 42 (22.1%) belonged to the vitamin D deficiency group, 79 (41.6%) belonged to the vitamin D insufficient group, and 69 (36.3%) belonged to the vitamin D sufficient group. In both intervention arms, none of the school children had potentially toxic levels (Table 2).

In the vitamin D supplementation arm, 232 (98.7%) had normal calcium levels and post-intervention, 230 (97.8%) had normal calcium levels. In the calcium intervention arm, all school students had normal calcium levels and 184 (96.8%) had normal calcium levels post-calcium supplementation (Table 3). 

In the vitamin D intervention arm and the calcium intervention arm, there was statistically significant improvement in Vitamin D levels (Table 4).

By comparing the Beck depression scores before and after the intervention, it was found that the vitamin D intervention arm showed a statistically significant reduction in Beck depression scores (Table 5).

By comparing the Beck depression scores post-intervention, it was found that the vitamin D arm showed a statistically significant reduction in Beck depression scores (Table 6).

Before the intervention, 235 adolescent students in the vitamin D intervention arm and 216 adolescent students in the calcium arm took part in the initial vitamin D and Beck depression evaluations. However, after two months of intervention, only 216 adolescent students in the vitamin D arm and 154 students in the calcium arm completed the vitamin D and Beck depression evaluations (Table 7).

The intention-to-treat analysis demonstrated that the vitamin D arm showed a statistically significant reduction in Beck depression scores (Table 8).

The difference-in-difference analysis showed a statistically significant difference in the BDI scores (Table 9).

Before intervention, the correlation between vitamin D levels and BDI scores indicated a very weak or no correlation that was found to be non-significant in the vitamin D arm (r = 0.04, *p* value = 0.5). After vitamin D intervention, there was weak but significant correlation observed between BDI scores and vitamin D levels (r = 0.139, *p* value = 0.03) (Figure 2).

## 4. Discussion

The present study was a cluster randomized controlled trial carried out for a period of three years, in which rural adolescent boys and girls were divided into an intervention arm that received a supplementary dose of 2250 IU of vitamin D and a non-intervention arm that received 250 IU per day with 500 g of calcium for a period of 9 weeks. The vitamin D intervention arm showed a statistically significant reduction in Beck depression scores compared with the calcium intervention arm. An intention-to-treat analysis showed that the vitamin D arm had a statistically significant reduction in Beck depression scores. The results from the present intervention trial suggest that vitamin D supplementation has a beneficial effect in reducing the total depression score obtained using the BDI-II questionnaire. There are scant studies on this domain of vitamin D supplementation and depression among adolescents. A study conducted by A. Bahrami et al. in Iran, wherein adolescents were supplemented with 50,000 IU of vitamin D for 9 weeks—a much higher dose than that in the present study—showed similar results of improvements in depression scores [18]. A meta-analysis conducted by F Vellekkatt and V Menon on the efficacy of vitamin D supplementation in the treatment of major depression showed that vitamin D supplementation favorably impacted depression ratings in individuals with major depression [19]. A study conducted by T. Jääskeläinen et al. suggested a protective effect of vitamin D against depressive disorder, showing that a higher serum 25(OH)D concentration is associated with a reduced risk of depressive disorder and depressive symptoms in a representative sample of the Finnish adult population [20]. A systematic review conducted by Shaffer et al. showed that vitamin D supplementation may be effective in reducing depressive symptoms in patients with clinically significant depression [21]. A meta-analysis confirmed the potential benefits of vitamin D supplementation and higher serum vitamin D levels in reducing the development and symptoms of depression, demonstrating its therapeutic role [22]. A systematic review conducted by Simon Spedding showed that, after adjusting for biological factors, vitamin D supplementation was efficacious in treating depression [23]. The present study administered a much lower dose of 2250 IU for 9 weeks to the vitamin D intervention arm. The meta-analysis results demonstrated that vitamin D has a beneficial impact on both the incidence and the prognosis of depression. Whether suffering from depression or not, individuals with low vitamin D levels are most likely to benefit from receiving a dose of >2800 IU as an intervention for a duration of ≥8 weeks. Possible reasons for this could be the functions of vitamin D, as it has a neuroprotective effect on the brain, and it possibly lowers plasma C-reactive protein levels in patients with psychiatric disorders and modulates inflammation by suppressing pro-inflammatory cytokines [24]. A meta-analysis showed that vitamin D level determination and vitamin D supplementation are affordable and safe. Thus, both actions could be good routine clinical practices in patients with mental illnesses such as depression, suicidal thoughts, and anxiety, as vitamin D enhances serotonin synthesis [25]. A cost-effective analysis of the national program of vitamin D supplementation among Iranian adolescents aged 11–18 years showed that vitamin D supplementation was effective in reducing adolescent depressive symptoms, as determined through cost savings and increases in QALYs compared to those who received no intervention, strongly suggesting that vitamin D supplementation may be a cost-effective strategy for decreasing depression symptoms in adolescents with a 100% probability [26]. Internalizing conditions, including depression and anxiety, can have pessimistic and gloomy effects on the wellbeing of children and adolescents. A dose higher than the recommended dietary allowance and a short duration of vitamin D supplementation appear to be safe and effective in reducing these internalizing conditions, suggesting that there needs to be a policy to encourage sufficient vitamin D intake for children and adolescents [27].

However, various studies have shown that vitamin D supplementation does not improve depression. An RCT conducted by Kjaergaard et al. showed that low levels of serum 25(OH)D are associated with depressive symptoms, but no effect was found with vitamin D supplementation [28]. A systematic review done by Guzek et al. showed that effectiveness of vitamin D supplementation in the treatment of depression still needs evidence generation through studies conducted at different part of the world and for now Vitamin D supplementation is still not beneficial in Depression treatment [29]. An RCT conducted by Jorde R and Kubiak L found that both the vitamin D and placebo groups showed a significant reduction in depressive symptoms, as evaluated via BDI-II scores. However, when comparing the changes in the two groups, no significant effect of vitamin D supplementation on BDI-II scores was found, even when the analyses were restricted to subjects with low serum 25(OH)D levels and mild depression [30]. A meta-analysis conducted by Gowda et al. did not find supporting evidence for the efficacy of vitamin D in improving depression among adults [31]. A study conducted by Mulugeta et al. in Australia showed that vitamin D may not help to prevent depression; however, the monitoring and treatment of vitamin D deficiency may be beneficial in alleviating the adverse influences of depression on health, contributing to a better quality of life in individuals with depression [32]. A randomized controlled trial found that vitamin D supplementation substantially increased 25(OH)D levels in patients with MDD and (severe) vitamin D deficiency, but did not materially improve antidepressant outcomes with flexibly dosed escitalopram; however, depressed patients with vitamin D deficiency may require higher antidepressant drug doses to experience benefits similar to those whose deficiency can be corrected by vitamin D supplementation [33].

The present study has many strengths: This is the first study of its kind, as there are very few studies examining vitamin D levels and depression in tropical countries such as India, and there are none in rural adolescents with the aim of helping to establish the causal relationship between vitamin D levels and depression in adolescents. All precautions were taken to avoid contamination between the experimental group and the control group. Sufficient time was allotted for interviews. Post-intervention follow-ups were adequately planned while looking for any adverse events. A limitation of the present study is the small sample size. Additionally, despite all precautions taken, attrition in the calcium arm could not be addressed, as some school children did not complete the interview, and some migrated during the trial. The long-term effects of vitamin D supplementation need to be assessed, as this was not carried out in the present study. Those students with higher scores for depression after Vitamin D and Calcium intervention were referred to a Psychiatrist for counselling; this would have been better with an initial clinical psychologist evaluation, with later referral to a Psychiatrist. 

## 5. Conclusions

Nutritional interventions widen the achievable treatment options for mental health problems among adolescents, as they are economical, easily accessible, more acceptable, and associated with fewer side effects when carried out appropriately, and they can be of extreme importance in many low- and middle-income countries (LMICs) where micronutrient deficiencies are common and supplementation may be a cost-effective public health intervention. The present randomized controlled trial shows that vitamin D supplementation improved depression scores among rural adolescents, stressing the importance of incorporating vitamin D supplementation into school health programs. In the future, studies with larger sample sizes and in different geographical locations should be carried out to examine the possible beneficial effects of vitamin D on adolescents with depression. 

## Figures and Tables

**Figure 1 nutrients-16-01828-f001:**
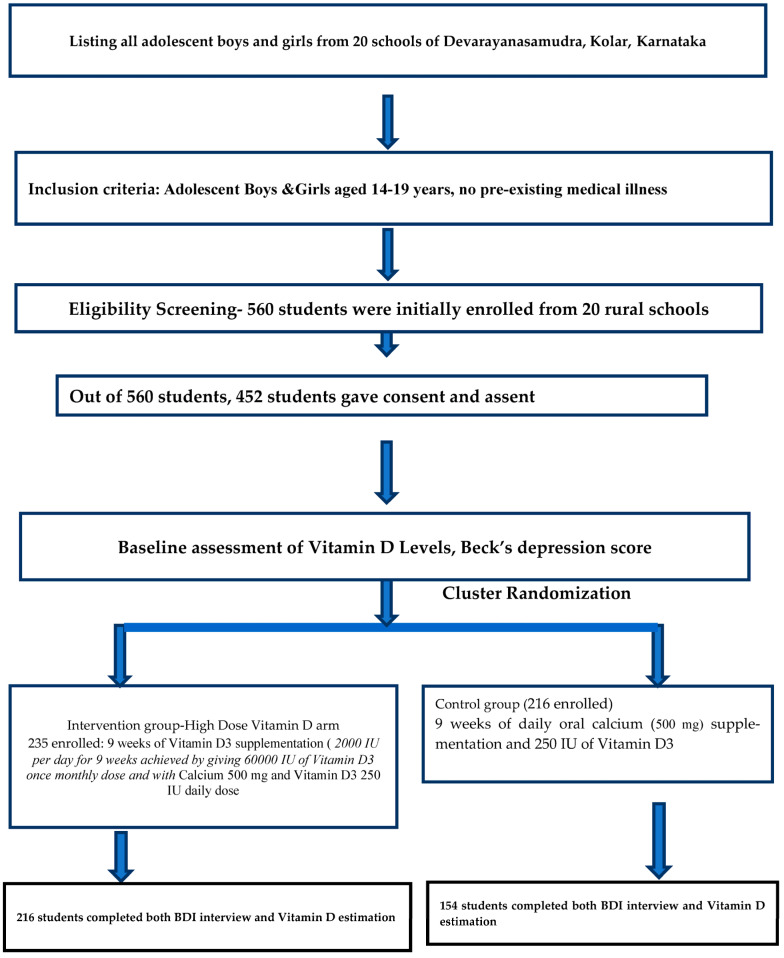
Flowchart depicting the participant’s recruitment and study procedure.

**Figure 2 nutrients-16-01828-f002:**
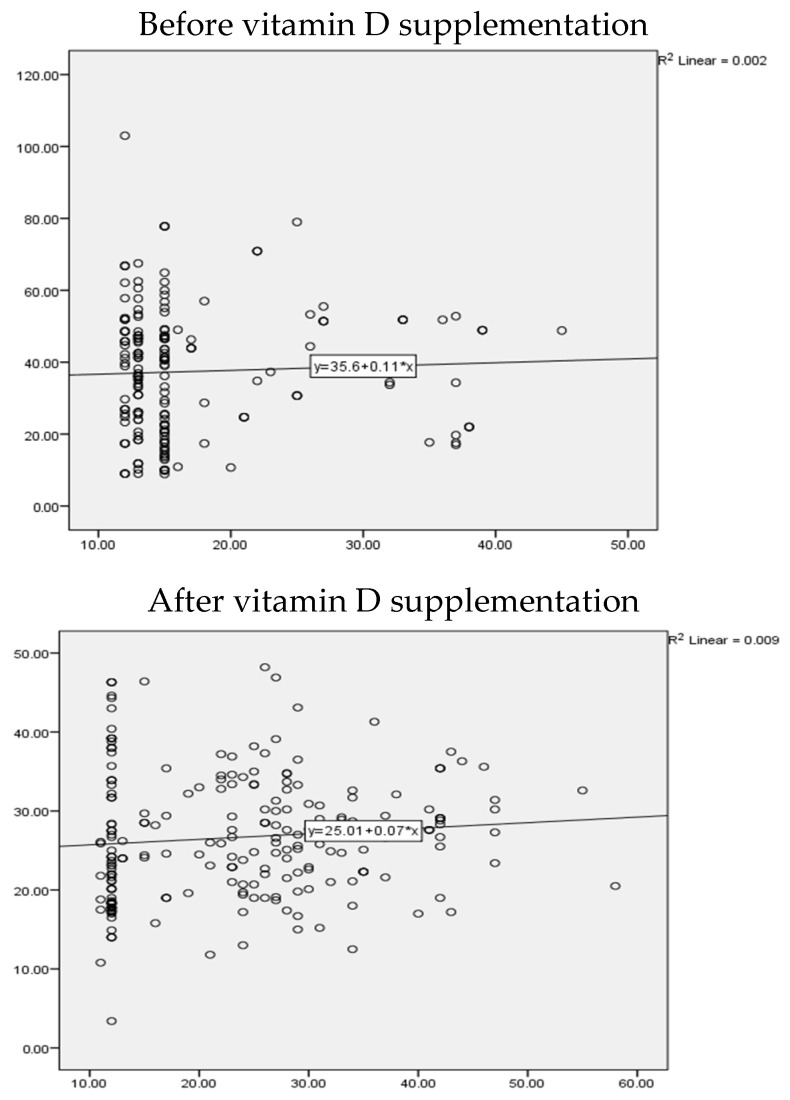
Scatter plot to correlate between depression scores and vitamin D levels in the vitamin D arm.

**Table 1 nutrients-16-01828-t001:** Distribution of adolescent school children according to sociodemographic profile.

Sociodemographic Profile		Vitamin D Supplementation Arm (n = 235)		Calcium Supplementation Arm (n = 216)	
		Frequency	Percent	Frequency	Percent
Age in years	14	8	3.4	42	19.4
	15	129	54.9	143	66.2
	16	86	36.6	31	14.4
	17	12	5.1	00	00
Gender	Boys	124	52.8	100	46.3
	Girls	111	47.2	116	53.7
Type of family	Nuclear	187	79.6	136	63.0
	Joint	48	20.4	80	37.0
Diet	Vegetarian	40	17.0	32	14.8
	Mixed	195	83.0	184	85.2
Timing of outdoor activities	Afternoon	166	70.6	56	25.9
	Evening	69	29.4	160	74.1
Duration of outdoor activities	Less than 30 min per day	51	21.7	105	48.6
	More than 30 min per day	184	78.3	111	51.4
Mean cluster size		25.6 ± 5.4		23.6 ± 9.4	

**Table 2 nutrients-16-01828-t002:** Distribution of adolescent school children according to vitamin D levels before and after intervention.

	Vitamin D Levels	Pre-Intervention		Post-Intervention	
		Frequency	Percent	Frequency	Percent
Vitamin D Intervention Arm	Deficient	53	22.6	54	23.0
	Insufficient	111	47.2	36	15.3
	Sufficient	71	30.2	145	61.7
	Total	235		235	100.0
Calcium Intervention Arm	Deficient	68	31.4	42	22.1
	Insufficient	59	27.3	79	41.6
	Sufficient	89	41.3	69	36.3
	Total	216	100.0	190	100.0

**Table 3 nutrients-16-01828-t003:** Distribution of adolescent school children according to calcium levels before and after intervention.

	Calcium Levels	Pre-Intervention		Post-Intervention	
		Frequency	Percent	Frequency	Percent
Vitamin D Intervention Arm	8.5–10.5	232	98.7	230	97.8
	More than 10.5	3	1.3	5	2.8
Calcium Intervention Arm	8.5–10.5	216	100.0	184	96.8
	More than 10.5	--	--	6	3.2

**Table 4 nutrients-16-01828-t004:** Comparison of serum vitamin D levels and serum calcium levels before and after intervention among adolescent students.

	Vitamin D Intervention Arm		Calcium Intervention Arm	
	Before Intervention	After Intervention	Before Intervention	After Intervention
Serum Vitamin D levels	26.6 ± 7.8	36.4 ± 17.7	28.3 ± 10.4	30.1 ± 9.8
*p* value	0.01		0.01	
Serum Calcium Levels	9.4 ± 0.4	9.7 ± 0.4	9.2 ± 0.4	9.3 ± 1.2
*p* value	0.01		0.45	

**Table 5 nutrients-16-01828-t005:** Comparison of Beck depression scores before and after intervention among adolescent students.

	Pre-Intervention BDI	Post-Intervention BDI	*p* Value #
Calcium Intervention Arm	21.5 ± 10.9	21.9 ± 12.5	0.59
Vitamin D Intervention Arm	20.4 ± 9.2	17.0 ± 7.3	0.001

# Paired *t*-test.

**Table 6 nutrients-16-01828-t006:** Comparison of Beck depression scores after intervention among adolescent students.

	Intervention	Mean ± Std. Deviation	*p* Value *
Beck Depression scores	Vitamin D Intervention Arm	17.0 ± 7.3	<0.01
	Calcium Intervention Arm	21.9 ± 12.5	

* Independent *t*-test.

**Table 7 nutrients-16-01828-t007:** Distribution of adolescent school children according to Beck depression classification before and after intervention.

		Before Intervention		After Intervention	
		Frequency	Percent	Frequency	Percent
Vitamin D Intervention Arm	Minimal	87	37.0	83	43.0
	Mild	17	7.2	16	8.3
	Moderate	64	27.2	60	31.1
	Severe	67	28.5	34	17.6
	Total	235	100.0	193	100.0
Calcium Intervention Arm	Minimal	81	37.5	64	41.6
	Mild	43	19.9	14	9.1
	Moderate	26	12.0	23	14.9
	Severe	66	30.6	53	34.4
	Total	216	100.0	154	100.0

**Table 8 nutrients-16-01828-t008:** Intention-to-treat analysis of BDI scores.

Intention-to-Treat Analysis	Vitamin D Arm	Calcium Arm
	Mean ± SD	Mean ± SD
Pre-Intervention BDI	20.4 ± 9.2	21.5 ± 10.9
Post-Intervention BDI	17.0 ± 7.3	21.9 ± 12.5
*p* Value	<0.001	0.54

**Table 9 nutrients-16-01828-t009:** Difference-in-difference analysis of BDI scores between the vitamin D arm and calcium arm.

DID	Intervention Arm	N	Mean ± Std. Deviation	*p* Value
BDI Scores	Vitamin D Arm	193	3.4 ± 8.3	0.01
	Calcium Arm	154	0.4 ± 11.4	

## Data Availability

The original contributions presented in the study are included in the article, further inquiries can be directed to the corresponding author.

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
