# Peer review of "Does Vitamin D3 Supplementation Improve Depression Scores among Rural Adolescents? A Randomized Controlled Trial"

_nutrients, 2024, doi:10.3390/nu16121828_

Round 1

Reviewer 1 Report

Comments and Suggestions for Authors

Dear Editor

The manuscript seems to consider a serious and interesting topic.

However, the manuscript requires significant editorial intervention.

1.       The manuscript must be proofread by a native speaker with a good medical (epidemiology) background.

2.       Use of capital letters is extensive and not fully justified.

3.       Abstract needs improvement (please use full stop after Background etc.). Conclusion needs improvement, please make it more scientific. Please, provide detailed information concerning Kolar, geographic region, country, longitude.

4.        Please, indicate how many children was included in the study (Material and Methods)

5.        Please, use nomenclature consequently (Vitamin D3 not Vitamin d or Vitamin D)

6.       Please, use the same units (ng/mL rather than ng/dl). Please do not use abbreviation 25-OH

7.       Did you take into consideration solar exposition and time of the year?

8.       Please provide the data concerning vitamin D and calcium levels in each group with standard deviations.

9.       Also, it seems that both arms were not fully randomized especially concerning mild and moderate according to Beck depression score.

10.   Please try to plot the depressing score against vitamin D3 level and observe possible correlation.

11.   In discussion please also consider that the change in vitamin D level from deficiency or severe deficiency (below 10 ng/mL) to sufficiency may have an impact on depression score. The effect of supplementation may not be strong in already sufficient patients.

Comments on the Quality of English Language

1. The manuscript must be proofread by a native speaker with a good medical (epidemiology) background.

2.       Use of capital letters is extensive and not fully justified.

Author Response

Comments 1: The manuscript must be proofread by a native speaker with a good medical (epidemiology) background.
Response 1: Thank you for pointing this out.  I/We agree with this comment.  

Comments 2: Use of capital letters is extensive and not fully justified.
Response 2: Agree with this comment.  Changes have been made almost everywhere where ever it was needed.
Comments 3: Abstract needs improvement (please use full stop after Background etc.).  Conclusion needs improvement, please make it more scientific.  Please, provide detailed information concerning Kolar, geographic region, country, longitude.
Changed as per suggestion.
Comments 4: Please, indicate how many children was included in the study (Material and Methods)
Response : Flow chart added for better understanding
Comments 5: Please, use nomenclature consequently (Vitamin D3 not Vitamin d or Vitamin D)
Response : Changes made everywhere
Comments 6: Please, use the same units (ng/mL rather than ng/dl).  Please do not use abbreviation 25-OH
Response: Changes made everywhere to ng/ml.  25-OH in VITROS Immunodiagnostic products with a 25-OH Total reagent pack is actually the name in the reagent pack.  So it is used.  We would like to stick to the same.
Comments 7: Did you take into consideration solar exposition and time of the year?
Response: The present study was done in same geographical locale so solar exposition was not taken in to account.  Time of year where study was done although varied in days among different schools it was done in early summer months of February and March.
Comments 8: Please provide the data concerning vitamin D and calcium levels in each group with standard deviations.
Response : Provided
Comments 9: Also, it seems that both arms were not fully randomized especially concerning mild and moderate according to Beck depression score.
Response: the present study randomized schools into clusters after baseline evaluation to avoid contamination.  No individual randomization was done.  Unit of randomization in the present study were schools and not individuals.
Comments 10: Please try to plot the depressing score against vitamin D3 level and observe possible correlation.
Comments 11: In discussion please also consider that the change in vitamin D level from deficiency or severe deficiency (below 10 ng/mL) to sufficiency may have an impact on depression score.  The effect of supplementation may not be strong in already sufficient patients.
Response: we accept this comment but the objective of the present study was to look for whether supplementation of Vitamin D reduces depression scores or not.  So we did not look into this during analysis.Although Table 7 gives some evidence regarding variation in frequency (percentage) among depression category before and after intervention in intervention arm. 

4.  Response to Comments on the Quality of English Language
Point 1:
Response 1:    Quality of English has been verified by MDPI referred

Reviewer 2 Report

Comments and Suggestions for Authors

The topic of the article is certainly interesting and noteworthy, above all I believe that it can be published with some corrections as a pilot study even if carried out on a fairly large number of topics.

However, there are several critical issues and I would also like to consider the following observations as suggestions for the possible continuation of the study-research.

1- Particularly in childhood and adolescence, as also described in the DSM-5 TR but not only, it is known that there are various types of depression and not only those considered clinically valid, therefore, obviously, it is not so simple to make a diagnosis of depression valid in this age group.

2- Another point concerns the age of the recruited sample. While in the materials and methods we talk about a very wide age range, 11-18 years, and therefore of subjects with very diversified neurodevelopment and psychological evolution situations, in the results, in tab. 1, the demographics for vitamin D and calcium supplementation are 14-17

3 -Regarding the instrument used, the BDI-II (1996) has international standardized cutoff values used that differ from the original: 0–13: minimal depression 14–19: mild depression 20–28: moderate depression 29–63: severe depression. This does not appear to be the one used by the authors.

4 - Finally, I would like to recommend using in the future, as I hope that the study can have a follow-up, given its positive peculiarities, another psychodiagnostic tool or, better yet, more psychological tools better if administered and analyzed by a qualified clinical psychologist . Among the various psychodiagnostic detection tools present in the literature, and specifically created for the developmental age and adolescence, there is also the DSS of which I attach the bibliography (Pruneti, C., Guidotti, S., 2021. Depressive states, behavioral and cognitive components in developmental age: factor analysis of a brief assessment instrument. Mediterranean Journal of Clinical Psychology, 9(1).https://doi.org/10.6092/2282-1619/mjcp-2842). In fact, I believe that limiting oneself to the sole evaluation of "depression yes and depression no", i.e. the presence of any type of symptomatology without dealing with the underlying balance and without evaluating the facets of the disorder could then also undermine the interesting data deriving from the integration intervention food.

Author Response

Comments 1: Particularly in childhood and adolescence, as also described in the DSM-5 TR but not only, it is known that there are various types of depression and not only those considered clinically valid, therefore, obviously, it is not so simple to make a diagnosis of depression valid in this age group.

Response 1: We accept this comment. Even though this is clearly stated, we as researchers find it very tricky to answer as  currently we do not have one definitive tool which can address this vexed question. 

Comments 2: Another point concerns the age of the recruited sample. While in the materials and methods we talk about a very wide age range, 11-18 years, and therefore of subjects with very diversified neurodevelopment and psychological evolution situations, in the results, in tab. 1, the demographics for vitamin D and calcium supplementation are 14-17

Response 2: Agree. I/We have, accordingly, done/revised/changed/modified…..to emphasize this point. The current study was although planned for adolescent boys and girls from age 11 to 19 years we faced few operational difficulties. First to describe was that 11 to 14 years many students provided us consent and assent but while withdrawing blood they had syncope issues which was creating a panic situation among rural school children and teachers compromising our study at those schools and second to say was we could make out that swallowing vitamin d and calcium tablets/ supplementation was very difficult as there was resistance from children as they found it difficult to swallow. The very question of adherence was difficult to achieve. And lastly many schools had children aged 18 and 19 years for which no consent was obtained from school principal citing that they are chronic failures and this intervention may impact their our academics.  

Comments 3: Regarding the instrument used, the BDI-II (1996) has international standardized cutoff values used that differ from the original: 0–13: minimal depression 14–19: mild depression 20–28: moderate depression 29–63: severe depression. This does not appear to be the one used by the authors

Comments 4: Finally, I would like to recommend using in the future, as I hope that the study can have a follow-up, given its positive peculiarities, another psychodiagnostic tool or, better yet, more psychological tools better if administered and analyzed by a qualified clinical psychologist . Among the various psychodiagnostic detection tools present in the literature, and specifically created for the developmental age and adolescence, there is also the DSS of which I attach the bibliography (Pruneti, C., Guidotti, S., 2021. Depressive states, behavioral and cognitive components in developmental age: factor analysis of a brief assessment instrument. Mediterranean Journal of Clinical Psychology, 9(1).https://doi.org/10.6092/2282-1619/mjcp-2842). In fact, I believe that limiting oneself to the sole evaluation of "depression yes and depression no", i.e. the presence of any type of symptomatology without dealing with the underlying balance and without evaluating the facets of the disorder could then also undermine the interesting data deriving from the integration intervention food.

Response : we cordially accept this recommendation. we will carry out future study with this valuable tool and identify newer dimensions of this most sorted mental problem. Thank you for your valuable suggestion/s.

4. Response to Comments on the Quality of English Language

Round 2

Reviewer 1 Report

Comments and Suggestions for Authors

Thank you for all corrections and clarifications

Author Response

Respected reviewers ,

we have only one corrections made for Revision 2. we have added the solar exposure status in the methodology.(same has been highlighted)

Rest all we have already done as asked before